# A Bayesian In-Play Prediction Model for Association Football Outcomes

**Qingrong Zou [1]**, **Kai Song [2,3]** and **Jian Shi [2,3,*]**

1   School of Applied Science, Beijing Information Science and Technology University, Beijing 100192, China; qingrongzou@bistu.edu.cn
2   Academy of Mathematics and Systems Science, Chinese Academy of Sciences, Beijing 100190, China; kaisong@amss.ac.cn
3   School of Mathematical Sciences, University of Chinese Academy of Sciences, Beijing 100049, China
*   Correspondence: jshi@iss.ac.cn

**Abstract:** Point process models have made a significant contribution to the prediction of football association outcomes. It is conventionally the case that defence and attack capabilities have been assumed to be constant during a match and estimated against the average performance of all other teams in history. Drawing upon a Bayesian method, this paper proposes a dynamic strength model which relaxes assumption of the constant teams' strengths and permits applying in-match performance information to calibrate them. An empirical study demonstrates that although the Bayesian model fails to achieve improvement in goal difference prediction, it registers clear achievements with regard to the prediction of the total number of goals and Win/Draw/Loss outcome prediction. When the Bayesian model bets against the SBOBet bookmaker, one of the most popular gaming companies among Asian handicaps fans, whose odds data were obtained from both the Win/Draw/Loss market and over–under market, it may obtain positive returns; this clearly contrasts with the process model with constant strengths, which fails to win money from the bookmaker.

**Keywords:** bayesian inference; point process model; in-play prediction; betting market; adjusting forecast; association football

## 1. Introduction

Association football is the most popular sport in the world and betting on outcomes has a long tradition. In addition, football constitutes the fastest growing gambling market [1]. As a result, the modelling and prediction of football match outcomes has become increasingly popular [2–4]. The prediction of an individual match outcome is a challenging task, which can be broken down into two types: prediction pre-match and prediction in-play.

Pregame prediction previously used the Negative Binomial model for association football scores over the Poisson model Reep [5]. However, since the seminal paper by Maher [6], the Poisson distribution goal-based data analysis, in which match results are generated by the two teams' attack and defence parameters, has been widely deployed [2,7–9]. The plausibility of Poisson regression models for the interpretation and prediction of association football (soccer) scores were examined in [10]. In addition, because the Poisson count model only applies in instances where the data are time-homogeneous and equi-dispersed, McShane [11] proposed the count model based on Weibull interarrival times, which were capable of managing both under and over-dispersed data. Boshnakov et al. [12] applied the Weibull model for forecasting association football scores, and the results demonstrated it performed better than the Poisson count model. Recently, Ref. [13] introduced a probabilistic model for forecasting whether the teams in a soccer competition will reach the target

that they set at the beginning of the championship. Ref. [14] developed a suitable probability model for studying the points achieved by a team in a football match. A substantial amount of literature has focused upon improving the aforementioned models, for example, Ref. [15] developed a hierarchical Bayesian Poisson model in which the scoring rates of the teams are convex combinations of parameters estimated from historical data and the additional source of the betting odds. Suzuki et al. [16] made use of the specialists' opinions and the FIFA ratings as prior information, and applied the Bayesian methodology for predicting match outcomes. Constantinou et al. [1] presented a Bayesian network model for predicting football outcomes with subjective variables which represented the factors that were important for prediction but historical data failed to capture; Owen [17] proposed Bayesian forecasting models to allow the strengths of each team to vary over time. Ref. [18] accounted for team strengths using the line-up of fielded players, a mixed effects model used to identify a player's true goal scoring ability. Ref. [19] presented a player-based model for football scores which takes into account the abilities of the players on each team. In addition, their basic model for the scoreline in a football match is a bivariate Weibull count model proposed by [12].

Live betting is very popular within betting markets and prediction in-play is therefore a significant factor that deserves closer consideration. However, among the huge amount of literature published on prediction of association football games, only a few papers concentrated on the in-play prediction. Dixon et al. [20] developed a pure birth process model, where the processes of goal times of home and away teams are taken to be two nonhomogeneous Poisson processes. In attempting to align themselves with practical circumstances, Zou et al. [21] proposed a discrete-time and finite-state Markov chain model that is grounded within the Poisson processes, where no more than one goal in a minute had happened, except during time intervals (44, 45] and (89, 90] in consideration of injury time, and a recursive algorithm was derived to accurately calculate the outcome probability. Volf [22] and Titman [23] studied the effects of other events such as cards in match. Volf [22] considered a semi-parametric model that contains a non-parametric baseline intensity, with the regression component reflecting the actual match state and defensive strength of rival teams. The authors in Titman [23] used a real-time eight-dimensional multivariate counting process to seek the interplay between football event processes—this did not only model the interdependence between home and away team goals; to the same extent, it sought to quantify the effect that cards had on the course of the outcome of a game. The results suggested the award of yellow cards did not appear to directly impact on goal scoring rates; in contrast, red cards had a substantial detrimental effect, in particular when the away team was reduced to ten men.

One major drawback of the pure birth process model is that it does not consider using in-match information to update the teams' strengths, which means that the teams' strengths are assumed to be constant as the match progresses. In actual fact, the birth process model, in making the prediction of the following score conditional on the current score at time $T$, only makes use of the score at time $T$, which enhances the prediction accuracy by reducing the transition (goal) times to the final score. In addition, the estimated strength parameters of team $i$ are based on the average performance against all other teams in history. While the model that contains other events can use in-match information to calibrate the scoring rate, the prediction of events often proves to be quite difficult.

In order to incorporate both historical match information and in-match information into an in-play prediction model, we propose a dynamic teams' strength calibration model that is based on the Bayesian method, which enables to apply in-match performance information to calibrate the estimates of each team's strengths. In addition, we achieve a prior estimation of team's strengths by using historical match information.

The rest of the paper is organized as follows. Section 2 describes the model for goal times before introducing the Bayesian inferences. Section 4 describes the data and the results of parameter estimations as well as the out-of-sample performance. Section 5 then describes betting strategies and betting results. In bringing the paper to a close, Section 6 draws conclusions and proposes further work.

## 2. Model for Goal Times

Before introducing how to update the teams' strength by the in-match information, we firstly summarize the model of Dixon and Robinson (1998). It is the basic model for the calibration of the teams' ability parameters, which is also named as the pure birth process model. The basic assumption of the model is that home-away scoring process is thought of as a two-dimensional inhomogeneous Poisson process. Considering the goal scoring process for a particular match $k$ between home team $H(k)$ and away team $A(k)$, there are two scoring processes, for goals of home team and away team with intensities $\Lambda_k(t)$ and $\Omega_k(t)$ that are allowed to vary with time $t$ and with status of the process. The intensity functions are

$$\Lambda_k(t) = a\alpha_{H(k)}\beta_{A(k)}\tau_{xy}(t)\rho(t) + \xi_1 t \tag{1}$$

and

$$\Omega_k(t) = \alpha_{A(k)}\beta_{H(k)}\kappa_{xy}(t)\rho(t) + \xi_2 t, \tag{2}$$

where $\alpha_{H(k)}$ measures the attack strength (the higher the value of $\alpha$, the stronger the attack) of home team $H(k)$; $\beta_{A(k)}$ is the defence strength (the smaller the value of the $\beta$, the stronger the defence) of away team $A(k)$; $a$ is the home advantage parameter; $\tau_{xy}$ and $\kappa_{xy}$ are parameters that determine the scoring rates during which the score is $(x, y)$; $t \in [0, 1]$ is the (rescaled) time elapsed during the match, and $\xi_1$ and $\xi_2$ reflect continuous variation within the rates over time (thereafter, the time $t$ in intensity functions is rescaled time). For league matches, the recoded match information is 90 min. Matches are played over two periods, each of 45 min. The time of treatment of the injury shall be made up in each half. $\rho(t)$ is used to model the injury time effect. As no data are available showing how much injury time is added, goal times of 45 and 90 min are considered as (possibly) censored observations. The parameters, that represent a multiplicative adjustment to the scoring intensity over the periods $(44, 45]$ and $(89, 90]$ min. are

$$\rho(t) = \begin{cases} \rho_1, & \text{if } t \in (44/90, 45/90]; \\ \rho_2, & \text{if } t \in (89/90, 90/90]; \\ 1, & \text{otherwise.} \end{cases} \tag{3}$$

Dixon and Robinson (1998) found in the best fitting-model the parameter $\tau_{xy}$ could be appropriately defined as

$$\tau_{xy}(t) = \begin{cases} \tau_{10}, & \text{if } x = 1, y = 0; \\ \tau_{01}, & \text{if } x = 0, y = 1; \\ \tau_{21}, & \text{if } x + y > 1, x - y \geq 1; \\ \tau_{12}, & \text{if } x + y > 1, x - y \leq -1; \\ 1, & \text{otherwise,} \end{cases} \tag{4}$$

and the parameter $\kappa_{xy}(t)$ could also be defined similarly, where the score is $(x, y)$ at time $(t-1)/90$.

The basic framework of inference is the likelihood funtion. For a particular match $k$, the likelihood for the process, in essence, is that of a two-dimensional non-homogeneous Poisson process, which can be derived by considering the process as a sequence of independent times between goals. With the independent increment of the process, if the total number of goals $m_k$ in match $k$ is more than 0, the likelihood takes the subsequent form:

$$L(t_k, J_k) = e^{-\int_0^1 \Lambda_k(t)dt - \int_0^1 \Omega_k(t)dt} \prod_{l=1}^{m_k} \Lambda_k(t_{k,l})^{1-J_{k,l}} \Omega_k(t_{k,l})^{J_{k,l}}, \tag{5}$$

if there is no goals, the likelihood takes the subsequent form:

$$L(t_k, J_k) = e^{-\int_0^1 \Lambda_k(t)dt - \int_0^1 \Omega_k(t)dt},$$ (6)

The observed data are $\{(t_{k,l}, J_{k,l}), 1 \leq l \leq m_k\}$, $t_{k,l}$ is the rescaled time of the $l$th goal and $J_{k,l}$ is an indicator which is 0 for a home goal and 1 for an away goal. In addition, Dixon and Robinson assumed scores in one match are independent of the scores in another match, so that the overall likelihood can be obtained by taking the product over matches.

In the model, we need to estimate $2d + 13$ parameters with $d$ teams, which is a high-dimensional nonlinear optimization problem. Then we use the coordinate descent algorithm to solve this problem. In addition, the model parameters $\alpha_i$, $\beta_j$, $\tau_{xy}$, $\kappa_{xy}$, $\rho_1$, $\rho_2$, $\xi_1$, $\xi_2$, $a$, $(i, j = 1, 2...d)$ are continually updated during each round—this is because in a round all teams appear and only appear once, with $d$ being the number of teams. To be more specific, we fit the model on the training dataset and forecast outcomes of the following round' matches. Having made predictions, we then enlarge the training dataset by taking into account the predicted matches and refit the model. We repeat this procedure until the last round of games have been forecasted. For forecasting, the recursive algorithm [21] is applied by us to calculate the outcome probability. The model coding and the following Bayesian inference are implemented in Matlab. Bayesian analysis and Markov chain Monte Carlo (MCMC) simulation are performed by referring [24,25].

## 3. Bayesian Inference

The main idea of this paper is using previous matches as prior information and then updating it through in-game information. The specific procedure of the Bayesian inference is illustrated in Figure 1. Except for teams' strength parameters, other model parameters are assigned prior distributions–degenerate distributions. In other words, these parameters are assumed to be known and constant for all teams throughout one round, and the values are equal to estimators with history matches based on the pure birth process model. In addition, for modelling the characteristics that are unique to the individual match, we firstly specify suitable distributions for teams' strength parameters and then calibrate them with observed in-match information, and we assume that means of prior distributions are equal to estimates based on the pure birth process model with historical matches.

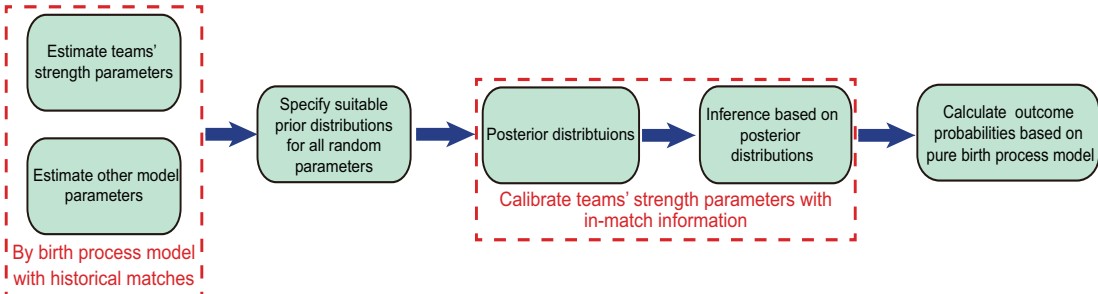

**Figure 1.** Illustration of the outcome prediction procedure. We estimate the parameters by the pure birth process model with the historical data, and use the Bayesian method to update the teams' strength parameters with the in-match information. Then the recursive algorithm is employed to calculate the outcome probability.

### 3.1. The Prior Distributions

For a specific match, from kick-off to time $T$ ($T \in 90$), if there are $m(m > 0)$ goals, the observed data is $Z = \{(t_l, J_l), 1 \leq l \leq m\}$, where the meaning of $(t_l, J_l)$ is in common with their meaning in Equation (5); if there are no goals, the observed data is $Z = \{(\infty, 0), (\infty, 1)\}$. For simplicity, we denote $X(T)$ and $Y(T)$ as the number of goals of home team and away team. According to the likelihood

function (5) and (6), the likelihood function is as follows: if the total goals are more than 0 from kick-off to time $T$, then

$$
\begin{aligned}
f(Z,\theta) =& e^{-\int_0^{T/90}(\lambda(t)+\xi_1 t)dt - \int_0^{T/90}(\mu(t)+\xi_2 t)dt} \prod_{l=1}^{m}(\lambda(t_l) + \xi_1 t_l)^{1-J_l}(\mu(t_l) + \xi_2 t_l)^{J_l} \\
\propto& e^{-\int_0^{T/90}\lambda(t)dt - \int_0^{T/90}\mu(t)dt} \prod_{l=1}^{m}(\lambda(t_l) + \xi_1 t_l)^{1-J_l}(\mu(t_l) + \xi_2 t_l)^{J_l},
\end{aligned}
\tag{7}
$$

if there are no goals from start to time $T$, then

$$
\begin{aligned}
f(Z,\theta) =& e^{-\int_0^{T/90}(\lambda(t)+\xi_1 t)dt - \int_0^{T/90}(\mu(t)+\xi_2 t)dt} \\
\propto& e^{-\int_0^{T/90}\lambda(t)dt - \int_0^{T/90}\mu(t)dt}
\end{aligned}
\tag{8}
$$

where $\lambda(t) = a\theta_1 \rho(t)\tau_{xy}(t)$, $\mu(t) = \theta_2 \rho(t)\kappa_{xy}(t)$. $\theta = (\theta_1, \theta_2)$, $\theta_1$ stands for the home team's scoring intensity parameter which is determined jointly by the attack and defence strength of the two team involved, $\theta_1 = \alpha_H \beta_A$. Similarly, $\theta_2$ stands for the away team's scoring intensity parameter, $\theta_2 = \alpha_A \beta_H$. It is worth noting that in above likelihood function, $(a, \rho(t), \tau_{xy}(t), \kappa_{xy}(t), \xi_1, \xi_2)$ is a constant vector.

Since the goal times are generally recorded to the integer part of the time of the goal, i.e., they are recorded in minutes, the integrals can be substituted with the closed form expressions:

$$
e^{-\int_0^{T/90}\lambda(t)dt - \int_0^{T/90}\mu(t)dt} = e^{-\frac{1}{90}\sum_{t=1/90}^{T/90}(\lambda(t)+\mu(t))}
\tag{9}
$$

Note that for the situation of total goals more than 0, if $\xi_1 = \xi_2 = 0$, $f(Z,\theta)$ is proportional to the probability mass function of Poisson distribution; if $\xi_1 \neq 0$ or $\xi_2 \neq 0$, $f(Z,\theta)$ is proportional to the sum of multiple probability mass functions of Poisson distributions. It is widely acknowledged that Gamma distribution is the conjugate prior distribution for Poisson distribution. For convenience of calculation, the prior distribution $\pi(\theta)$ is assumed to be formed by two independent Gamma distributions for home team and away team where the correlation between them is reflected by $\tau_{xy}$ and $\kappa_{xy}$. In addition, the copula is a common used method to examine the association or dependence between variables. For example, a copula very simple, and almost conjugated is provided in [26]. Ref. [12] used a Weibull inter-arrival-times- based count process and a copula to produce a bivariate distribution of the numbers of goals acored by the home and away teams in a match.

Moreover, we assume the mean of prior distribution of $\theta_1$ is $\hat{\theta}_{01}$ and the mean of prior distribution of $\theta_2$ is $\hat{\theta}_{02}$, where $\hat{\theta}_{01}$ and $\hat{\theta}_{02}$ are maximum likelihood estimations of $\theta_1$ and $\theta_2$ based on the pure birth process model with all historical matches.

To put it differently, the prior distributions for $\theta_1$ and $\theta_2$ are respectively $\Gamma(\omega_1, r_1)$ and $\Gamma(\omega_2, r_2)$, with $r_1$ and $r_2$ being the shape parameters and $\omega_1$ and $\omega_2$ operating as scale parameters. In addition, $r_1/\omega_1 = \hat{\theta}_{01}, r_2/\omega_2 = \hat{\theta}_{02}$. Then, the prior distribution of $(\theta_1, \theta_2)$ is

$$
\pi(\theta_1, \theta_2) = \pi(\theta_1)\pi(\theta_2) \propto e^{-\omega_1 \theta_1}\theta_1^{r_1-1} \cdot e^{-\omega_2 \theta_2}\theta_2^{r_2-1}.
\tag{10}
$$

### 3.2. The Posterior Distributions

Bayesian inferences are conditional on the actual data and analyses directly extrapolate from the posterior distribution, which provides prior and current in-match information on the parameters.

When the scoring information from start to time $T$ is observed, the posterior distribution of $\theta = (\theta_1, \theta_2)$ is

$$
\pi(\theta|Z) = \frac{\pi(\theta)f(Z,\theta)}{\int \pi(\theta)f(Z,\theta)d\theta} \propto \pi(\theta)f(Z,\theta).
\tag{11}
$$

If total goals are more than 0, the posterior distribution is in proportion to

$$
\begin{aligned}
\pi(\theta|Z) \propto & e^{-(a\frac{1}{90}\sum\limits_{t=1/90}^{T/90}\rho(t)\tau_{xy}(t)+\omega_1)\theta_1 - (\frac{1}{90}\sum\limits_{t=1/90}^{T/90}\rho(t)\kappa_{xy}(t)+\omega_2)\theta_2} \cdot \theta_1^{r_1-1}\theta_2^{r_2-1} \\
& \times \prod_{l=1}^{m} (a\rho(t_l)\tau_{xy}(t_l)\theta_1 + \xi_1 t_l)^{1-J_l}(\rho(t_l)\kappa_{xy}(t_l)\theta_2 + \xi_2 t_l)^{J_l};
\end{aligned}
\tag{12}
$$

if there are no goals, namely, $X(T) = Y(T) = 0$, the posterior distribution is in proportion to

$$
\begin{aligned}
\pi(\theta|Z) \propto & e^{-(a\frac{1}{90}\sum\limits_{t=1/90}^{T/90}\rho(t)\tau_{xy}(t)+\omega_1)\theta_1 - (\frac{1}{90}\sum\limits_{t=1/90}^{T/90}\rho(t)\kappa_{xy}(t)+\omega_2)\theta_2} \cdot \theta_1^{r_1-1}\theta_2^{r_2-1} \\
= & e^{-(a\frac{1}{90}\sum\limits_{t=1/90}^{T/90}\rho(t)\tau_{xy}(t)+\omega_1)\theta_1 - (\frac{1}{90}\sum\limits_{t=1/90}^{T/90}\rho(t)\kappa_{xy}(t)+\omega_2)\theta_2} \cdot \theta_1^{r_1+X(T)-1}\theta_2^{r_2+Y(T)-1}
\end{aligned}
\tag{13}
$$

It will be noted that when there are no goals, the posterior mean can be obtained in a concise form. However, when the number of total goals is more than zero, the posterior mean is obtained in a closed, albeit relatively complex form. In addition, when further Bayesian inference is performed, it is still necessary to use the Metropolis algorithm to generate samples from the posterior distribution. However, if we let $\xi_1 = \xi_2 = 0$, $f(Z, \theta)$ is in proportion to the probability mass function of Poisson distribution, and then the posterior distribution will be

$$
\pi(\theta|Z) \propto e^{-(a\frac{1}{90}\sum\limits_{t=1/90}^{T/90}\rho(t)\tau_{xy}(t)+\omega_1)\theta_1 - (\frac{1}{90}\sum\limits_{t=1/90}^{T/90}\rho(t)\kappa_{xy}(t)+\omega_2)\theta_2} \cdot \theta_1^{r_1+X(T)-1}\theta_2^{r_2+Y(T)-1}
\tag{14}
$$

which is Gamma distribution. Namely, the poster distributions of $\theta_1$ and $\theta_2$ are

$$
\theta_1|Z \sim Gamma(\frac{a}{90}\sum_{t=1/90}^{T/90}\rho(t)\tau_{xy}(t) + \omega_1, r_1 + X(T))
\tag{15}
$$

and

$$
\theta_2|Z \sim Gamma(\frac{1}{90}\sum_{t=1/90}^{T/90}\rho(t)\kappa_{xy}(t) + \omega_2, r_2 + Y(T)).
\tag{16}
$$

The estimates of parameters $\xi_1$ and $\xi_2$ are fairly small, which are usually less than 0.5. Take $\xi = 0.5$ for example, $\xi$ makes $\frac{1}{2}\xi t^2$ contributions to the expected number of goals, where $t \in (0, 1)$. For instance, when we have observed in-match information of the first half, if we did not consider the effect of $t$, the time of left to play, we would merely underestimate the expected number of goals by 0.06. In addition, empirical study shows that if the time left to play is not considered in the likelihood function, it has little effect on the final prediction accuracy. When total goals are more than 0, for likelihood function $f(Z, \theta)$, we use the following approximation

$$
f(Z, \theta) \propto e^{-\frac{1}{90}\sum\limits_{t=1/90}^{T/90}(\lambda(t)+\mu(t))}\prod_{l=1}^{m}\lambda(t_l)^{1-J_l}\mu(t_l)^{J_l}.
\tag{17}
$$

So no matter whether there are goals during $(0, T]$, the posterior distributions of $\theta_1$ and $\theta_2$ are shown in Equations (15) and (16) respectively.

In this paper, we use posterior means as estimations of teams' strengths calibrated with in-match information. With the posterior distributions (14) and (15), the posterior mean estimations are

$$
\hat{\theta}_1 = \frac{r_1 + X(T)}{r_1 + E_H(T)}\hat{\theta}_{01}
\tag{18}
$$

and

$$\hat{\theta}_2 = \frac{r_2 + Y(T)}{r_2 + E_A(T)}\hat{\theta}_{02}, \tag{19}$$

respectively. In the preceding estimations,

$$E_H(T) = \frac{a\hat{\theta}_{01}}{90} \sum_{t-1/90}^{T/90} \tau_{xy}(t)\rho(t) \tag{20}$$

and

$$E_A(T) = \frac{\hat{\theta}_{02}}{90} \sum_{t-1/90}^{T/90} \kappa_{xy}(t)\rho(t) \tag{21}$$

indicate the expected number of goals in $[0, T]$ for the home and away teams. We can see that the calculation of quantities $E_H(T)$ and $E_A(T)$ needs an integral over $\tau_{xy}(t)$ and $\kappa_{xy}(t)$, which depend on the random values of $X(t-1)$ and $Y(t-1)$, $t \in (0, T]$. To actually compute the expectations it would be necessary to consider the transition probabilities of the underlying paired birth process. In other words, we need to predict the outcome probabilities at time $T$ by using birth process model and recursive algorithm [21].

### 3.3. The Choice of $r_1$ and $r_2$

To add to our understanding further, taking home team for example, we can see that

$$\hat{\theta}_1 = \hat{\theta}_{01} + \frac{X(T) - E_H(T)}{r_1 + E_H(T)}\hat{\theta}_{01}. \tag{22}$$

Then the change rate of the posterior estimation relative to the prior estimation is $\frac{|\hat{\theta}_1 - \hat{\theta}_{01}|}{\hat{\theta}_{01}} = \frac{|X(T) - E_H(T)|}{E_H(T) + r_1}$, and $\frac{|X(T) - E_H(T)|}{E_H(T)}$ is the change rate of actual goals relative to the expected number of goals. If $0 < r_1 \leq E_H(T)$, the change rate will fall in $[\frac{1}{2}\frac{|X(T) - E_H(T)|}{E_H(T)}, \frac{|X(T) - E_H(T)|}{E_H(T)})$. If $E_H(T) < r_1 < \infty$, the estimate change rate will fall in $(0, \frac{1}{2}\frac{|X(T) - E_H(T)|}{E_H(T)})$. The lower the $r_1$, the greater the requirement to calibrate strengths with the in-match information. This makes it possible to identify that the shape parameter $r_1$ of prior distribution determines the balance between the influences of historical matches information ('the prior') and the in-match goal information.

There are several choices for the shape parameters of prior distributions. One common choice is specifying the variance, namely, adding the following conditions, $r_1/\omega_1^2 = \sigma_{\hat{\theta}_{01}}^2$ for home team and $r_2/\omega_2^2 = \sigma_{\hat{\theta}_{02}}^2$ for away team in a match, where $\sigma_{\hat{\theta}_{01}}^2$ and $\sigma_{\hat{\theta}_{02}}^2$ are variances of $\hat{\theta}_{01}$ and $\hat{\theta}_{02}$. Combining with $r_1/\omega_1 = \hat{\theta}_{01}$ and $r_2/\omega_2 = \hat{\theta}_{02}$, we obtain $r_1 = \hat{\theta}_{01}^2/\sigma_{\hat{\theta}_{01}}^2$ and $r_2 = \hat{\theta}_{02}^2/\sigma_{\hat{\theta}_{02}}^2$. However, because variances are quite small, the values of $r_1$ and $r_2$ are almost in $(30, 150)$—this suggests estimate change rates are quite close to zero and lead to no improvement.

Another common option is to search $r_1$ and $r_2$ to maximize the marginal distribution of the observed data –this is known as the empirical Bayes approach. In empirical studies, estimation results show that $r_h$ for home team is about 3 while $r_a$ for away team is about 5. However, in a match the average expectation goals for home team and away team are respectively only about 1.6 and 1.2. The preceding analysis suggests that, throughout the match, the estimate change rate will be lower than half of the change rate of actual goals relative to expected goals, especially for away teams. By applying the preceding two options, it becomes possible to identify that the model will put too much emphasis on the prior information to ignore the observed data.

The focus will now shift to search the shape parameters of prior distributions with the intention of balancing the effects of prior information and the newly observed match information. As the game proceeds, the level of in-match information will increase. It is therefore preferable to slightly calibrate strengths in the first half and to intensify the calibration of these strengths in the second half.

The preceding discussion suggests that an appropriate value of $r_1$ may be $E_H(45)$, which is specific for each match so that it can capture the presence of different quality in the teams and avoid producing over-shrinkage [27]. Because the expected number of goals $E_H(T)$ increases as match time progresses, the strength estimation change rate will fall in $(0, \frac{1}{2}\frac{|X(T)-E_H(T)|}{E_H(T)})$ during the first half, and the strength estimation change rate will fall in $(\frac{1}{2}\frac{|X(T)-E_H(T)|}{E_H(T)}, \frac{|X(T)-E_H(T)|}{E_H(T)})$ during the second half. The away parameter $r_2$ can be similarly analysed, and a proper value of $r_2$ can be set as $E_A(45)$.

In summary, for a single game, we choose expected goals of home team and away team at half-time as values of $r_1$ and $r_2$, namely, $r_1 = E_H(45)$, $r_2 = E_A(45)$.

## 4. Results

### 4.1. Data

We obtained goal time data of the English Premier League for the eight seasons from 2009/2010 to 2016/2017 from OPTA (https://www.whoscored.com/). Moreover, we also collected the live betting prices from OPTA of the SBOBet bookmaker. The betting data contain information on the over–under market, point-spread market and outcome (home win,draw,away win) betting market. We obtained 950 games' odds data from August 2013 to May 2015 and from January 2017 to May 2017.

### 4.2. Parameter Estimates

In this section we discuss the estimates of the parameters. Table 1 shows the estimates and standard errors of parameters which are assumed degenerated. The estimates are obtained based on the pure birth process model with all matches, and the standard errors are estimated by the observed Fisher information matrix.

**Table 1.** Estimates of the model parameters, based on the eight season matches. Standard errors are presented in parentheses.

| Parameter | Estimate (s.e.) | Parameter | Estimate (s.e.) |
|-----------|-----------------|-----------|-----------------|
| $\tau_{10}$ | 0.9793 (0.0423) | $\kappa_{10}$ | 1.1213 (0.0586) |
| $\tau_{01}$ | 1.0403 (0.0579) | $\kappa_{01}$ | 1.0818 (0.0588) |
| $\tau_{21}$ | 0.9175 (0.0389) | $\kappa_{21}$ | 1.0994 (0.0609) |
| $\tau_{12}$ | 1.1449 (0.0696) | $\kappa_{12}$ | 1.0417 (0.0599) |
| $\xi_1$ | 0.5851 (0.0390) | $\xi_2$ | 0.4028 (0.0333) |
| $\rho_1$ | 4.6335 (0.2759) | $\rho_2$ | 7.7464 (0.3585) |
| $a$ | 1.3824 (0.0243) | | |

In order to see the performance of the posterior distributions of the stochastic parameters, the confidence intervals are analysed. According to Equations (3) and (4), the confidence intervals for $\theta_1$ and $\theta_2$ are $[\frac{\chi 2(2V_H, \frac{\alpha}{2})}{2U_H}, \frac{\chi 2(2V_H, 1-\frac{\alpha}{2})}{2U_H}]$ and $[\frac{\chi 2(2V_A, \frac{\alpha}{2})}{2U_A}, \frac{\chi 2(2V_A, 1-\frac{\alpha}{2})}{2U_A}]$, respectively, where $U_H = a\frac{1}{90}\sum_{t=1/90}^{T/90}\rho(t)\tau_{xy}(t) + \omega_1$, $V_H = r_1 + X(T)$, $U_A = \frac{1}{90}\sum_{t=1/90}^{T/90}\rho(t)\kappa_{xy}(t) + \omega_2$, $V_A = r_2 + Y(T)$. Then we calculate the proportion of matches, whose prior estimates fall outside the confidence interval, to all matches. Although match time $t$ is continuous, goal times are generally recorded in minutes. Hence for per match, there are 89 time points, which are from 1 min to 89 min, we can use in-match information to calibrate teams' strength, so that there are $2 \times 89 = 178$ confidence intervals.

With 1520 matches from August 2013 to May 2017, there are 44.80% matches whose prior estimates of strength parameters fall outside at least one of the confidence intervals at 5% significance level; and 90.39% at 10% significance level. However, the above calculation may be a bit overvalued in that the prior estimates fall outside only one of the confidence interval may not indicate the teams' strengths are changed. Then for per match, we only use in-match information from kick-off to time

$T$ ($T$ = 5, 10, ..., 85) to calibrate teams' strength, so that there are $2 \times 17 = 34$ confidence intervals. In this case, there are 41.97% matches whose prior estimates of strength parameters fall outside the confidence interval at 5% significance level; and 86.84% at 10% significance level. These results provide encouraging signs about the model's validity and usefulness.

*4.3. Model Fit*

As explained in [24], once we have accomplished the first two steps of a Bayesian analysis–constructing a probability model and computing the posterior distribution of all estimands–we should assess the fit of the model to the data and to our substantive knowledge. The fundamental tool designed for achieving the task is posterior predictive checking. Its basic technique is to draw simulated values from the joint posterior predictive distribution of replicated data and compare these samples to the observed data. Any systematic differences between the simulations and the data indicate potential failings of the model.

We measure the discrepancy between model and data by defining test statistic $T(y)$. Lack of fit of the model with respect to the posterior predictive distribution may be measured by the tail-area probability, or $p$-value, of the test statistic, and computed using posterior simulations of $(\theta, y^{rep})$. Here, to avoid confusion with the observed data, $y$, we define $y^{rep}$ as the replicated data that might have been observed. If we already have $S$ simulations from the posterior density of $\theta$, we just draw one $y^{rep}$ from the predictive distribution for each simulated $\theta$; we now have $S$ draws from the joint posterior distribution, $p(y^{rep}, \theta|y)$. The posterior predictive check is the comparison between the realized test quantities, $T(y, \theta^s)$, and the predictive test quantities, $T(y^{reps}, \theta^s)$. For ordered discrete data we can compute a 'mid' $p$-value

$$p = Pr(T(y^{rep}) < T(y)|y) + \frac{1}{2}Pr(T(y^{rep}) = T(y)|y). \tag{23}$$

From an interpretative point of view, an extreme $p$-value–too close to 0 or 1–suggests a lack of fit of the model compared to the observed data, and a reasonable range of the $p$-value is between 0.05 and 0.95.

Specifically, we perform the posterior predictive test using the test quantity $T$ = the difference between home team's goals and away team's goals. For each match, we make 1000 simulations from the posterior density of $\theta$, and we have 1000 draws from the joint posterior distribution, $p(y^{rep}, \theta|y)$. Hence, an estimate for the Bayesian $p$-value is given by Equation (23). Figure 2 displays the box-plot of 1520 matches' $p$-values. The horizontal axis shows the scores in time interval $[0, T]$ are observed. In other words, the final score is predicted conditional on score at time $T$ ($T = 10, 20, \ldots, 70$). Further, in order to demonstrate more information, we add 95% and 80% confidence limits to the box-plot. The reported confidence intervals are based on sample quantile with 1520 matches' $p$-values. From this plot the fit of the model seems good–the replicated data under the model are plausible and close to the data at hand.

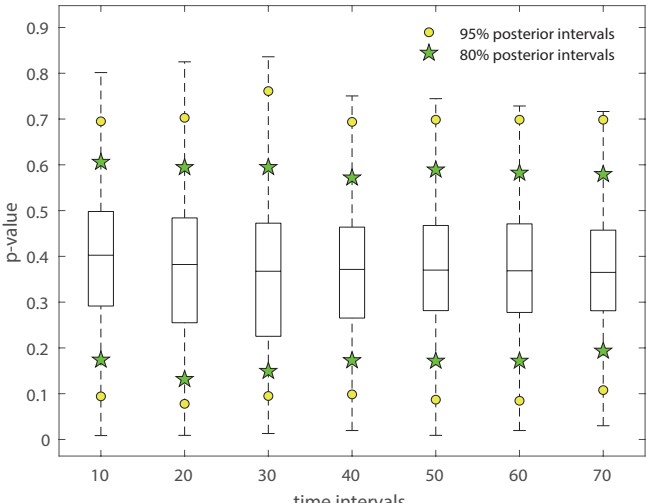

**Figure 2.** Box-plot of posterior predictive check for the goals' difference against the replicated goals' difference. In addition, we add the 95% and 80% confidence limits to the box-plot. The yellow-filled circles represent 95% confidence limits; the green-filled stars present 80% confidence limits.

*4.4. Out-of-Sample Performance*

4.4.1. Rank Probability Score (RPS)

In order to measure out-of-sample predictive accuracy, we compare our model against other models. Specifically, we use the the ranked probability score for Win/Draw/Loss outcomes. The Brier score (BS) and the ranked probability score (RPS) are widely used measures to describe the quality of categorical probabilistic forecasts. The BS can be regarded as the special case of an RPS with two forecast categories [28]. The RPS is particularly appropriate for evaluating probability forecasts of ordered variables [29,30] explained that RPS was the most rational scoring rule of those that have been proposed and used for football outcomes. For a single forecast the RPS is defined as

$$RPS = \frac{1}{s-1} \sum_{i=1}^{s-1} (\sum_{j=1}^{i} (p_j - e_j))^2 \tag{24}$$

where $s$ is the number of potential outcomes, and $p_j$ and $e_j$ are the forecasts and observed outcomes at position $j$. A lower score indicates a more accurate forecast (lower error).

We compare the RPS of our model with two other models, a pre-match forecasting model and an in-play forecasting model. Comparing with pre-match forecasting model, we might truly see whether in-match events contain additional information. A player based model proposed by [19], which is one of the state of the art model, is chosen as a comparator pre-match forecasting model. Their basic model for the scoreline in a football match is a bivariate Weibull count model described by [12]. In addition, the dynamic nature of team strengths are also incorporated into the player-based model. For in-play forecasting comparator model, our basic model is chosen, that is the pure birth process model.

In order to compare with the RPS results given in [19], the same test set, one and a half season's data (570 matches) starting from 2014–2015 season to 2015–2016 season is chosen. The best result of the player-based model is 0.2020, and Table 2 presents the RPS values of our model and the pure birth process model, which prediction probabilities are on condition of scores at time $T$. At kick-off, actually our model is the pure birth process model in that there is no in-match information to update teams' strengths. When information just from kick-off to 5 min is observed, the RPS values of our model and the pure birth process model are higher than that of the player-based model; the RPS value of our

model is lower than that of the pure birth process model. When information of the first 10 min after kick-off is obtained, the value of the Bayesian model is lower than that of the player-based model, however, the value of the pure birth process model is still higher than that of the player-based model. From 20 min after kick-off, both in-match models perform better than the pre-match model. We can see that as more and more information is observed, the RPS values are getting smaller and smaller for the period of prediction getting shorter. So the results of later part of the match can not explain whether in-match events contain additional information. However, the results of first 20 min may illustrate that applying in-match information to calibrate strengths is useful.

**Table 2.** The RPS values for two models, the prediction probabilities are conditional on scores at time $T$.

| $T$ | 0 | 5 | 10 | 20 | 45 | 70 |
|---|---|---|---|---|---|---|
| Approximation Bayesian model | 0.2210 | 0.2030 | 0.1983 | 0.1966 | 0.1529 | 0.0942 |
| Pure birth process model | 0.2210 | 0.2149 | 0.2093 | 0.1987 | 0.1512 | 0.0942 |

### 4.4.2. Calibration Curve

Calibration can be intuitively seen as a way to visualise how often a model is right or wrong [12]. In this section, we directly evaluate the calibration of the approximation Bayesian model's posterior prediction distribution using 1520 matches from from August 2013 to May 2017. For every prediction event, we visualise the model's performance graphically by plotting the calibration curve. Then we briefly describe how to estimate the calibration curve in football suggested by [12].

We divide the prediction space by 'halves': we split the data into upper and lower halves, then split those halves, then split the extreme halves recursively. Compared to equal-width bins, this allows intuitive visual inspection of tail behaviour. When the calibration curve lies below the diagonal, the model is optimistic in that it over-estimates the probability of the occurrence of the event; When the calibration curve lies above the diagonal, the model is pessimistic in that it under-estimates the probability of the occurrence of the event.

Figure 3 illustrates the calibration curve for forecasting home team win, draw and away team win. Subgraphs from top to bottom describe the curve for forecasting home team win, away team win and draw separately; subgraphs from left to right present empirical frequency versus model prediction probability conditional on score information at 10 min, 30 min and 50 min respectively. Although tail behaviour of both model is poor, overall it appears that our model is better calibrated than the pure birth process model. Moreover, as more score information is observed, the model is better calibrated.

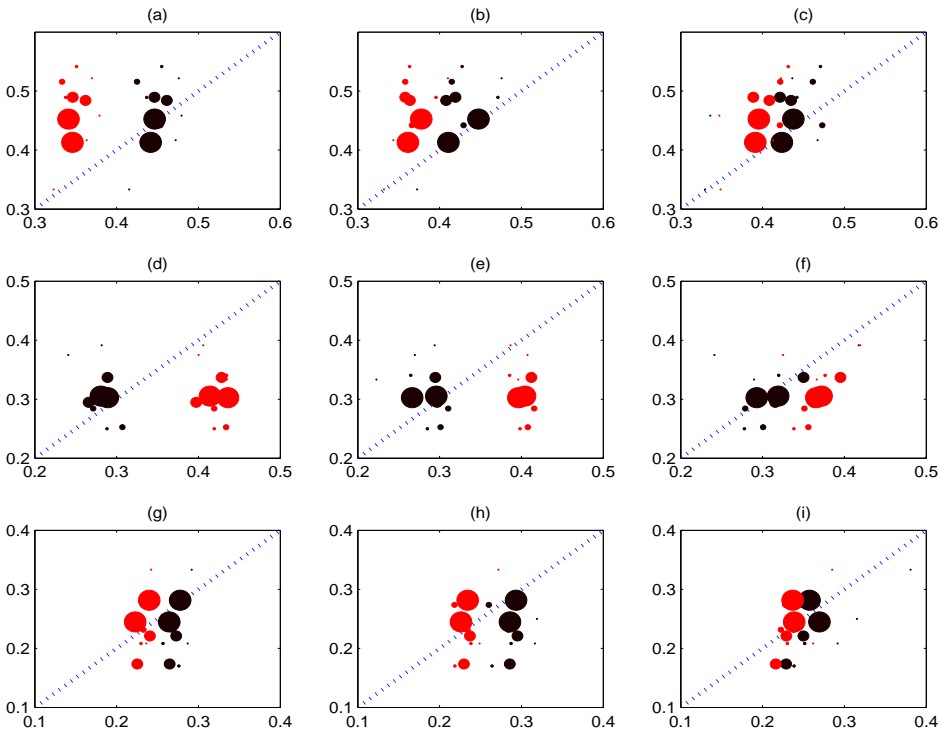

**Figure 3.** Calibration curve for predicting outcomes in home-draw-away market. The black filled circles represent our model's calibration curve and the red filled circles represent the pure birth process model's calibration curve. The size of the circles is proportional to the number of observations in each bin. The blue dot line represents the $y = x$ line. The horizontal axis shows model prediction probability, and the vertical axis represents the empirical frequency. The first row shows calibration curve for home team win; the middle row describes calibration curve for away team win; the bottom row exhibits calibration for draw. Subgraph (**a**,**d**,**g**) show the model prediction probability on condition of score information at 10 min versus empirical frequency; the middle column (subgraph (**b**,**e**,**h**)) presents the model prediction probability conditional on score information at 30 min versus empirical frequency. The right column (subgraph (**c**,**f**,**i**)) illustrates the model prediction probability conditional on score information at 50 min versus empirical frequency.

## 5. Application to In-Play Betting

### 5.1. Betting Strategy

The three aforementioned common markets are renowned for providing opportunities to bet on football matches. In the point-spread market, if a bettor bets on the favourite, he or she wins the bet if the favourite wins by more than the point spread. If a bettor bets on the underdog, he or she wins the bet if the underdog either wins, or loses by less than the point spread. If the difference in points is equal to the point spread, the bettor receives his wager back and the bet is effectively cancelled (known as a 'push'). Betting on the over–under market is analogous to the point-spread market, where the bettor can place money on more or fewer points being scored in the game than the figure identified by the bookmaker (the details are shown in the Appendix A).

In order to demonstrate the obtained return mainly originates from the model, we test the models using a simple one unit betting strategy: if the expected value of the bet was in excess of some value, then we bet one unit. To be more specific, for event type $A$, we only bet if

$$P(A) \times Odds(A) - 1 > \tau, \tag{25}$$

where $P(A)$ and $Odds(A)$ are the prediction probability and betting price of event $A$, $\tau$ is the threshold parameter. Increasing $\tau$ leads to a stricter betting regime, but consequently produces fewer bets. One unit will be staked when the aforementioned condition is satisfied. On the over–under and point-spread markets, no more than a single event will satisfy the betting condition; however, there may be more than one events meeting the betting condition on outcome betting market. When more than one events meet the condition, we only bet on the event with the highest expectation return. The strategy has also been applied in other papers contributed by Boshnakov et al. [12], Dixon et al. [7] and Koopman [9].

*5.2. Betting Performance*

In order to further verify the out-of-sample performance of our model in real-time prediction capability (e.g., during a match), we made bets every five minutes with the intention of calculating the average return. Bets were accordingly made at the following time points: 5, 10, 15, 20, 25, 30, 35, 40, 45, 50, 55, 60, 65, 70, 75, 80, 85.

The actual return of bets during the 2013/8–2015/5 and 2016/8–2017/5 seasons can be determined with reference to a range of $\tau$-values. Vigorish is standard in betting markets: if the bookmakers are accurate in their probability specifications, they will have an in-built 'take' that corresponds to their expected profit. To win money from bookmakers, in the sense of gaining a positive expected return, requires a determination of probabilities which are sufficiently more accurate than those obtained from the odds so as to overcome the bookmakers' take.

In Figure 4 we present the returns of betting on over–under market for various values of $\tau$. Both models' average returns are presented as the full curves and are compared with the negative average return of 3.04%, the bookmaker's take. The detailed mean return calculation procedure is as follows. Firstly, for a specific match, there are 17 time points that we need to consider whether to bet one unit for per time point according to Equation (25). Then we can obtain the total net return and total staked for a match, further, total net return and bet of all matches obtained. Finally, total net return divided by total bet of all matches is the mean return.

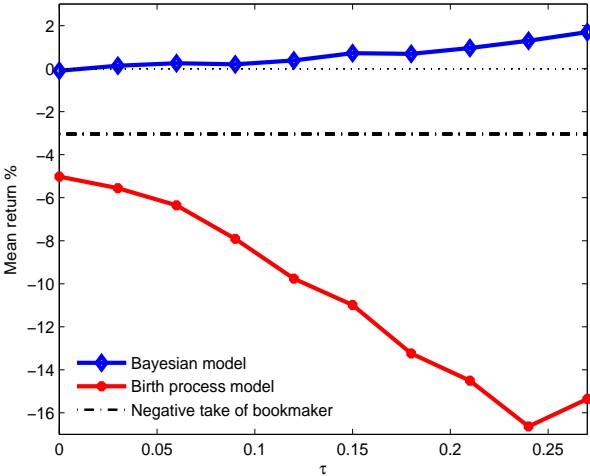

**Figure 4.** Average return from betting on over–under market for various values of the threshold $\tau$.

It will be noted that our model under the common strategy does not only achieve a return that exceeds −3.04% but also generates a positive return, upon the condition that threshold $\tau$ exceeds zero. However, the returns of the birth process model are considerably less than −3.04%.

As $\tau$ increases, with the concomitant establishment of a stricter betting regime, the returns of our model also increase while the returns of the birth process model decrease. One reason for returns decreasing as $\tau$ increases may be that more and more small probability events with very high odds are

chosen to be staked, with the consequence that we calculated the average betting right rate. The results demonstrate that, as $\tau$ increases, our model's average betting right rate only decreases from about 48% to 47%; however, this can be directly contrasted with the average betting right rate for the birth process model, which dropped from about 46% to 36%—this in turn verifies the anticipated results.

Using the same simple betting strategy, our model gains positive returns while the birth process model does not produce positive returns, which suggests that the returns mainly emanate from the model.

Figure 5 demonstrates the returns that derive from betting on the point-spread market for various values of $\tau$. The average returns of both model are presented as full curves and they are compared with the negative average return of 2.47% of the bookmaker's take. It will be noted that both model cannot gain positive returns. With small threshold values, our model performs better than the birth process model, however, both model perform worse than the bookmaker. With large threshold values, our model performs worse than the birth process model, and both models perform better than the bookmaker. The above phenomena implies that our model fail to enhance point spread prediction accuracy.

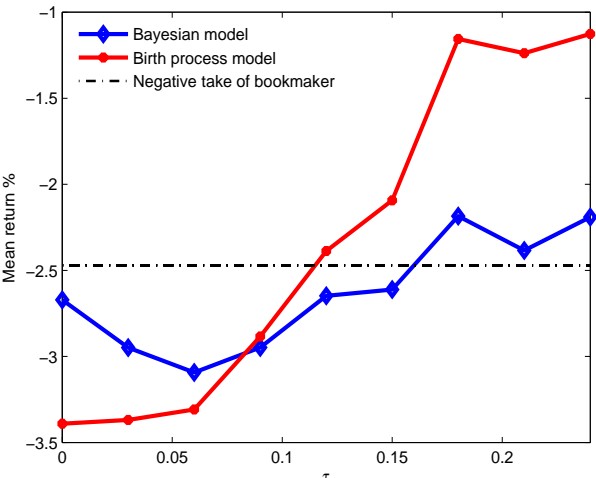

**Figure 5.** Average return from betting on point-spread market for various values of the threshold $\tau$.

Figure 6 describes the returns that derive from betting on match outcomes for various values of $\tau$. Average returns of both model are presented as the full curves and them are compared against the negative bookmaker's take 5.11%. It is immediately obvious that our model obtains returns that far exceed the average negative bookmaker's take $-5.11\%$. In addition, our model begins to obtain positive returns when the threshold $\tau$ exceeds 0.06—here it is noticeable that the returns increase as the threshold increases. For the birth process model, we can see that it cannot achieve positive returns. So our model has made a large improvement in the win,draw and loss prediction.

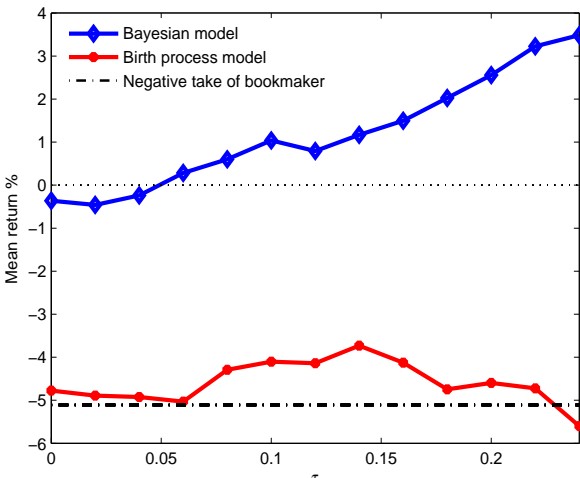

**Figure 6.** Average return from betting on outcome market for various values of the threshold $\tau$.

## 6. Conclusions and Future Work

In this paper, on the basis of Bayesian method, we propose a dynamic strength model which relaxes assumption of the constant teams' strengths and permits applying in-match performance information to calibrate teams' strengths. We test our model against the birth process model where teams' strengths are assumed to be constant during the match based upon betting performance on three common markets. With the intention of comprehensively verifying the performance for a match, we calculated the average return by considering betting every 5 min. The results demonstrate that our model can achieve a positive return and significantly outperform the birth process model on the over–under market. This extends to the outcome betting market when the threshold exceeds 0.06—in this circumstance our model succeeds in obtaining positive returns and therefore outperforms the birth process model (which also cannot achieve positive returns under different threshold values). However, applied to the point-spread market, our model does not clearly improve the forecasting of the score difference. However, this defect should be considered against the fact that our forecasting method has a clear and advantageous application to over–under market and Win/Draw/Loss betting market.

Although we have presented some promising results for our dynamic calibration strengths model with in-match information, we believe that further improvements can be envisaged and put into effect. First, it is important to acknowledge that the model failed to improve prediction accuracy for goal difference—this suggests that future engagements should attempt to develop a model focused upon goal difference. Second, the present model only uses in-match goal times information; diversification beyond this limited point to engage with separate event information such as red cards could contribute clear research benefits. One possible approach is to introduce covariates into the baseline strength parameters $\alpha$ and $\beta$, i.e., using covariates to describe the team strength. Furthermore, the important covariates can be selected by using Bayesian model choice. The relevant literature can be referred to in [31,32].

**Author Contributions:** Conceptualization, Q.Z. and J.S.; methodology, Q.Z., K.S. and J.S.; software, Q.Z.; validation, Q.Z., K.S. and J.S.; formal analysis, Q.Z.; investigation, Q.Z.; resources, Q.Z.; data curation, Q.Z. and J.S.; writing–original draft preparation, Q.Z.; writing–review and editing, K.S. and J.S.; visualization, Q.Z.; supervision, J.S.; project administration, J.S.; funding acquisition, J.S. All authors have read and agreed to the published version of the manuscript.

**Funding:** This research was funded by National Major Project grant number 2017ZX06002006.

**Acknowledgments:** We would like to thank the guest editors and reviewers for their valuable comments and suggestions which helped us to improve this paper. We would also thank Beijing StatusWin Lottery Operations Technology Ltd for providing the data.

**Conflicts of Interest:** The authors declare no conflict of interest.

**Appendix A**

In the appendix, Table A1 shows rules of Asian point-spread market, and the more detailed information can be referenced by https://en.wikipedia.org/wiki/Asian_handicap. Table A2 shows rules of Asian over–under market.

**Table A1.** Rules of Asian spread market.

| Handicap | Team Result | Bet Result | Handicap | Team Result | Bet Result |
|----------|-------------|------------|----------|-------------|------------|
| 0 | Win<br>Draw<br>Lose | Win<br>Stake Refund<br>Lose | 0 | Win<br>Draw<br>Lose | Win<br>Stake Refund<br>Lose |
| $-(N + 0.25)$ | Win by (N + 1)+<br>Win by N<br>Otherwise | Win<br>Lose half<br>Lose | $+(N + 0.25)$ | Lose by (N + 1)+<br>Lose by N<br>Otherwise | Lose<br>Win half<br>Win |
| $-(N + 0.50)$ | Win by (N + 1)+<br>Otherwise | Win<br>Lose | $+(N + 0.50)$ | Lose by (N + 1)+<br>Otherwise | Lose<br>Win |
| $-(N + 0.75)$ | Win by (N + 2)+<br>Win by N + 1<br>Otherwise | Win<br>Gain half<br>Lose | $+(N + 0.75)$ | Lose by (N + 2)+<br>Lose by N + 1<br>Otherwise | Lose<br>Lose half<br>Win |
| $-(N + 1.00)$ | Win by (N + 2)+<br>Win by N + 1<br>Otherwise | Win<br>Stake Refund<br>Lose | $+(N + 1.00)$ | Lose by (N + 2)+<br>Lose by N + 1<br>Otherwise | Lose<br>Stake Refund<br>Win |

[1] The negative handicap denotes home team conceding the ball and the positive handicaps denote the home team is transferee. [2] The plus signs in the Team result columns indicate 'or more', e.g., '(N + 1)+' means 'by N + 1 goals or more'.

**Table A2.** Rules of Asian over–under market.

| Handicap | Number of Goals | Profit for Over | Profit for Under |
|----------|-----------------|-----------------|------------------|
| N | N<br>No more than N − 1<br>No less than N + 1 | Stake Refund<br>Loss<br>Win | Stake Refund<br>Win<br>Loss |
| N + 0.25 | No less than N + 1<br>N<br>No more than N − 1 | Win<br>Lose half<br>Loss | Loss<br>Win half<br>Win |
| N + 0.50 | No less than N + 1<br>No more than N | Win<br>Loss | Loss<br>Win |
| N + 0.75 | No more than N<br>N + 1<br>No less than N + 2 | Loss<br>Win half<br>Win | Win<br>Lose half<br>Loss |

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
