# Peer review of "A Bayesian In-Play Prediction Model for Association Football Outcomes"

_applsci, doi:10.3390/app10082904_

Round 1
Reviewer 1 Report
I think this is a competently executed and interesting paper. Bayesian analysis is based on the Gamma distribution. I have no objections to the methods other than the following comment: It is a good idea to substitute all integrals with closed form expressions related to what the authors actually used in approximating them in discrete time.
Author Response
Thank you for this valuable feedback. We have substitute integrals related to our works with closed form expressions. (Please see page 5, line 148-149 and eq. (9); page 6, eq. (12)-eq. (17); page 7, eq. (20)- eq. (21); page 8, line 251)
Reviewer 2 Report
The paper is very interesting for a practical and theoretical points of view. The authors propose a dynamic strength model which relaxes assumption of the constant teams’ strengths and permits applying in-match performance information to calibrate them, using a Bayesian inference scheme. More specifically, they employ point process models. I have some suggestion for improving the quality of the manuscript. See below.
- Please clarify how do you set the values “44/90” and/or “89/90” in the equation at page regarding \rho(t).
- Please, add more description in the caption of Figure 1.
- Please, add numbers to the equations. It is difficult to refer to your equations without numbers.
- You prior distribution are quite complex. What is the reason? model conjugacy? please clarify.
- Are you computing the marginal likelihood (Bayesian model evidence) for model selection? i.e., the denominator of the first equation of Section 3.2. Please, discuss it at least as future works considering the excellent survey papers
P. Congdon. Bayesian model choice based on Monte Carlo estimates of posterior model probabilities. Computational statistics and data analysis, 50(2):346–357, 200
F. Llorente, L. Martino, D. Delgado, J. Lopez-Santiago, "Marginal likelihood computation for model selection and hypothesis testing: an extensive review", viXra:2001.0052, 2019.
You can discuss it in in introduction and/or conclusions as future work.
- Regarding the inference: are using Monte Carlo, MCMC or importance sampling methods. Please, clarify this point adding an additional discussion. For instance, are you using Metropolis or advanced Metropolis algorithms (such as multiple try Metropolis)? or are you using adaptive importance sampling etc. Please, cite some related surveys in this discussion. This discussion can increase the number of interested readers and improve the impact of your work.
- Please, if your work is accepted, upload it in Research Gate and ArXiv in order to help its diffusion.​
Reviewer 3 Report
In this article Bayesian methodology is used to propose and study the relationship between strength and attack of soccer teams. This study is of considerable interest not only for the team owners themselves and their main actors, the players, fans etc. but also from the economic point of view due to the importance that the sports betting business plays today. In general, the work is very well written and moderately well presented (especially the graphics), it has an immediate interest and obviously fits perfectly with the present journal. I have no objection that if the authors try to respond to the suggestions stated below, it can be accepted for publication.
\section*{Comments}
\begin{enumerate}
\item Please define what the SBOBet bookmaker means in the abstract.
\item In Section 2 rewrite the phrase "Before describing our model for updating teams' strengths via in-match information, we summarize our basic model-the model of Dixon and Robinson (1998)(It is also referred to as pure birth process model)."
\item It is not clear to me what the role of the $\rho (t)$ parameter is. It should be explained in a little more detail. For example, why limit yourself to the value 90 when in many football matches (in most of them) this time limit is exceeded. For example, consider a final with an extension.
\item The authors literally point out that "distribution. For convenience of calculation,
141 the prior distribution p (q) is assumed to be formed by two independent Gamma distributions ". That is, independence is assumed because mathematically (and computationally) it is much easier. In practice it seems hardly assumable. This work points out something about it. For example, the introduction of a copula would seem a more reasonable modeling. I do not mean here that the work is introduced and redone, but that the authors say something about it. For example, a copula very simple, and almost conjugated is provided in \cite{lee_1996}.
\item Page 5, line 56. The authors should show how the posteior means are, and more if they say that they can be obtained in a simple way (I get a little lost, but I think they are given later). One of the advantages of the Bayesian proposal consists in the interesting interpretation of the a posteriori means in terms of the a priori information and the sample information (as long as the posterior mean is linear in relation to the data).
\item Starting on page 6 of the document there are many mathematical expressions that appear written between the lines. It would be better if they appear as a separate equation. That is, use the eqnarray environment.
\item Is there a way to introduce covariates in the proposed modeling that allows me to further refine the model?
\item I find the text that appears in the conclusions quite praiseworthy, literally stating "First, it is important to acknowledge that the model failed to improve prediction accuracy for goal
411 difference - this suggests that future engagements should attempt to develop a model focused upon
412 goal difference. Second, the present model only uses in-match goal times information - diversification
413 beyond this limited point to engage with separate event information such as red cards could contribute
414 clear research benefits ".
The authors have a proposal in this regard to save these situations.
\item Finally, I believe that the authors have ignored some recently published works in the literature and that are shown in the References Section of this report. Some of them are \cite{karlisandntzoufras_2000}, \cite{karlisandntzoufras_2003} and the recent papers provided by \cite{gomezdavilaperez_2019} and \cite{gomezanddavila_2019}.
\end{enumerate}
